# Being Underweight Increases the Risk of Non-Cystic Fibrosis Bronchiectasis in the Young Population: A Nationwide Population-Based Study

**DOI:** 10.3390/nu13093206

**Published:** 2021-09-15

**Authors:** Bumhee Yang, Kyungdo Han, Sang Hyuk Kim, Dong-Hwa Lee, Sang Hyun Park, Jung Eun Yoo, Dong Wook Shin, Hayoung Choi, Hyun Lee

**Affiliations:** 1Department of Medicine, Division of Pulmonary and Critical Care Medicine, Chungbuk National University Hospital, Chungbuk National University College of Medicine, Cheongju 28644, Korea; ybhworld0415@gmail.com; 2Department of Statistics and Actuarial Science, Soongsil University, Seoul 06978, Korea; hkd917@naver.com (K.H.); ujk8774@naver.com (S.H.P.); 3Samsung Medical Center, Department of Medicine, Division of Pulmonology and Critical Care Medicine, Sungkyunkwan University School of Medicine, Seoul 06351, Korea; gost702@naver.com; 4Department of Internal Medicine, Division of Endocrinology and Metabolism, Chungbuk National University Hospital, Cheongju 28644, Korea; roroko@hanmail.net; 5Healthcare System Gangnam Center, Department of Family Medicine, Seoul National University Hospital, Seoul 03080, Korea; ujungeun@gmail.com; 6Samsung Medical Center, Department of Family Medicine, Sungkyunkwan University School of Medicine, Seoul 06351, Korea; dwshin.md@gmail.com; 7Samsung Advanced Institute for Health Sciences & Technology (SAIHST), Sungkyunkwan University, Seoul 06351, Korea; 8Department of Internal Medicine, Division of Pulmonary, Allergy and Critical Care Medicine, Hallym University Kangnam Sacred Heart Hospital, Hallym University College of Medicine, Seoul 07441, Korea; 9Department of Internal Medicine, Division of Pulmonary Medicine and Allergy, Hanyang University College of Medicine, Seoul 04763, Korea

**Keywords:** bronchiectasis, body mass index, underweight, risk factors, young adults

## Abstract

Although body mass index (BMI) is a potential risk factor for bronchiectasis in young adults, the association between BMI and incident bronchiectasis has not been well elucidated. This study included 6,329,838 individuals aged 20–40 years from the Korean National Health Insurance Service database 2009–2012 who were followed up until the date of the diagnosis of bronchiectasis, death, or 31 December 2018. We evaluated the incidence and risk of bronchiectasis according to the BMI category. The incidence rate of bronchiectasis increased as BMI decreased in a dose-dependent manner (*p* for trend <0.01). In multivariable Cox regression analysis, being underweight was an independent risk factor for the development of bronchiectasis, with a hazard ratio of 1.24 (95% confidence interval, 1.19–1.30) compared to being normal weight. In subgroup analysis, the effect of being underweight on the development of bronchiectasis was more evident in males and older individuals (30–40 years) than females and younger individuals (20–29 years), respectively (*p* for interaction <0.01 for both). These results remained significant in subgroup analysis in which subjects with comorbidities related to being underweight were excluded. Being underweight may be a novel risk factor for the development of bronchiectasis in young adults.

## 1. Introduction

Non-cystic fibrosis bronchiectasis (hereafter referred to as bronchiectasis) is a chronic lung disease characterized by chronic respiratory symptoms and recurrent infection [1]. The disease has been getting more attention because the worldwide prevalence of bronchiectasis has been increasing in recent years [2,3]. Furthermore, the disease burden of bronchiectasis, including medical costs and mortality, is substantial [4,5]. Early recognition and interventions to prevent disease progression in young adults could potentially decrease the disease burden of bronchiectasis.

Current guidelines recommend an extensive work-up to evaluate the etiology of bronchiectasis in young adults. Identifiable etiologies in young adults include immunodeficiency, autoimmune disease, and primary ciliary dyskinesia, while respiratory infections are thought to be the leading cause of the disease [6,7]. Despite the importance of underlying conditions in the development of bronchiectasis in young adults, a significant proportion of patients have idiopathic bronchiectasis. Thus, factors that contribute to bronchiectasis, including demographic factors such as nutritional status, require further exploration [8].

Of several nutritional indicators, body mass index (BMI) is the most commonly used clinical parameter in the field of chronic respiratory diseases, including bronchiectasis [9,10]. The association between BMI and bronchiectasis has been relatively well evaluated, especially in terms of disease severity and prognosis [10,11]. However, although patients with bronchiectasis have lower BMI than those without bronchiectasis [12,13], it is unclear whether lower BMI is a risk factor of bronchiectasis. Therefore, this study aimed to investigate the effect of BMI on the development of bronchiectasis in young adults by analyzing data from a nationwide cohort of young adults.

## 2. Methods

### 2.1. Study Population and Design

We used the Korean National Health Insurance Service (NHIS) database [14]. South Korea has a single-payer universal health system; the NHIS maintains claims data on all reimbursed inpatient and outpatient visits, procedures, and prescriptions. Additionally, the NHIS database includes data from annual or biennial health screening exams provided free of charge by the Ministry of Health and Welfare. Approximately 72% of all eligible persons underwent screening in 2011–2014 [15].

This study initially included 6,861,282 adults aged 20–39 years who participated in a health screening exam between 1 January 2009 and 31 December 2012. We excluded subjects who had missing information for at least one variable (n = 489,173), those diagnosed with bronchiectasis before the enrollment period (n = 35,329), and those diagnosed with cystic fibrosis (n = 382), and those who died within one year of enrollment (n = 6560). Finally, a total of 6,329,838 subjects were included in this study (Figure 1). The cohort was followed from baseline to the date of bronchiectasis diagnosis, death, or the end of the study period (31 December 2018), whichever came first.

The study protocol was approved by the Institutional Review Board of Chungbuk National University Hospital (No. 2021-01-026). The requirement for informed consent was waived because the NHIS database was constructed after the anonymization of patient data.

### 2.2. Exposure: BMI

BMI was calculated as weight in kilograms divided by height in meters squared, and categorized according to Asian-specific criteria: underweight (<18.5 kg/m^2^), normal weight (18.5–22.9 kg/m^2^), overweight (23–24.9 kg/m^2^), obese (25.0–29.9 kg/m^2^), and severely obese (≥30 kg/m^2^) [16].

### 2.3. Outcome: Bronchiectasis

The main study outcome was the incidence of bronchiectasis. Bronchiectasis was defined by claims under the International Statistical Classification of Diseases and Related Health Problems, 10th revision (ICD-10) diagnosis code J47 (bronchiectasis) without a concomitant diagnosis of cystic fibrosis (E84), which was used in our previous study [3].

### 2.4. Covariates

Smoking status was determined by a self-administered questionnaire during the health screening exam, and subjects were categorized as either never-, ex-, or current smokers. Income level was dichotomized at the lowest 20%; the low-income category also included Medicaid beneficiaries. Data on alcohol consumption and regular physical activity were collected from self-administered questionnaires. Categories for alcohol consumption were none (0 g/day), mild (<30 g/day), and heavy (≥30 g/day). Categories for exercise were regular (>30 min of moderate physical at least 5 times per week or >20 min of strenuous physical activity at least 3 times per week) and non-regular.

Comorbidities that can influence the risk of bronchiectasis were also defined using the following ICD-10 codes: asthma (J45–J46), pulmonary tuberculosis (TB) (A15–A19), non-tuberculous mycobacteria (NTM) (A31) infection, diabetes mellitus (DM) (E10–E14), chronic kidney disease (CKD) (N18.1–N18.5 and N18.9), gastroesophageal reflux disease (GERD) (K21.0), solid cancer (C00–C97), connective tissue disease (M05, M06, M32, M35 and M45), hematologic malignancy (C90, C910, C920, C921, C922, C924–926, C928 and C930), transplantation (Z940, Z944, Z941 and Z942), human immunodeficiency virus infection (HIV) and acquired immune deficiency syndrome (AIDS) (B20–B24), and inflammatory bowel disease (K50–K51) [3,5].

### 2.5. Statistical Analysis

Baseline characteristics of participants are presented as means (standard deviations (SD)) or numbers (%) according to the BMI category. The incidence rate of bronchiectasis was calculated by dividing the number of events by 100,000 person-years (PY). Cox proportional hazards regression analyses were conducted to obtain the hazard ratios (HRs) and 95% confidence intervals (CIs) for the occurrence of bronchiectasis based on BMI category. The risk of bronchiectasis development was analyzed before and after adjustment for potential confounding factors. Model 1 was unadjusted. Model 2 was adjusted for age, sex, smoking status, alcohol consumption, regular exercise, low income, area of residence (rural or urban), and the number of hospital visits. Model 3 was additionally adjusted for respiratory disease, connective tissue disease, solid cancer, hematologic malignancy, transplantation, immunodeficiency, inflammatory bowel disease, and HIV and AIDS. Stratified analysis was performed by stratifying subjects according to age (<30 years and ≥30 years) and sex.

A subgroup analysis was further performed after excluding subjects with comorbidities potentially associated with low BMI (respiratory disease, connective tissue disease, solid cancer, hematologic malignancy, transplantation, immunodeficiency, inflammatory bowel disease, and HIV and AIDS) to minimize the effect of these comorbidities on the development of bronchiectasis since low BMI might have been caused by these comorbidities.

Statistical analyses were conducted using SAS software (version 9.4; SAS Institute, Cary, NC, USA), and statistical significance was defined as a two-sided *p*-value < 0.05.

## 3. Results

### 3.1. Baseline Characteristics

Of 6,329,838 participants, the mean age was 30.9 ± 5.0 years and 59.3% were men. During the study period, the mean numbers of outpatient visits and hospitalizations were 3.6 ± 6.5 and 0 ± 0.3, respectively. Among all study subjects, the major comorbidities were GERD (10.7%), asthma (5.2%), and DM (2.0%) (Table 1). Table 1 also depicts the baseline characteristics of the study population according to BMI. The underweight category comprised the youngest population and had the highest proportion of females across BMI categories. As BMI decreased, the proportions of never-smokers, non-alcohol drinkers, and low-income subjects increased, while the proportions of regular exercisers and rural residents decreased. The underweight category had the highest number of outpatient visits across BMI categories. Regarding comorbidities, the underweight group had the highest proportions of GERD, respiratory diseases, solid cancer, and inflammatory bowel disease, and the lowest proportion of DM compared with other BMI categories.

### 3.2. BMI and Risk of Bronchiectasis

During a mean follow-up of 7.4 years, 23,804 (0.4%) subjects developed bronchiectasis. The incidence rate of bronchiectasis increased as BMI decreased (*p* for trend <0.01) in both unadjusted and adjusted models. In the fully adjusted model, the risk of bronchiectasis development was significantly higher in the underweight category (HR 1.36, 95% CI 1.30–1.42) and lower in the overweight (HR 0.83, 95% CI 0.80–0.86), obesity (HR 0.82, 95% CI 0.79–0.85), and severe obesity categories (HR 0.79, 95% CI 0.73–0.84) compared with the normal BMI category (Table 2). The cumulative incidence of bronchiectasis according to the BMI category is shown in Figure 2.

### 3.3. Stratified Analyses by Sex and Age Group

Similar trends were consistently found in subgroup analyses stratified by sex and age group (<30 years vs. ≥30 years) (*p* for trend <0.01 for both). BMI category had a significant interaction with both sex and age group in terms of bronchiectasis development (*p* for interaction <0.01 for both). The effect of being underweight on bronchiectasis development was more significant in males than in females and in older (30–39 years) rather than younger (20–29 years) individuals (Table 2), which also mirrored the cumulative incidence of bronchiectasis when stratified by sex and age (Figure 3).

### 3.4. Subgroup Analysis Excluding Individuals with Comorbidities Potentially Caused by Being Underweight

We further investigated the relationship between BMI and the risk of bronchiectasis by excluding 518,955 individuals with comorbidities potentially associated with low BMI to minimize the effect of these comorbidities on the development of bronchiectasis. As shown in Table 3, the underweight category (HR 1.35, 95% CI 1.29–1.42) still had a higher risk of bronchiectasis development than the normal weight population; in contrast, overweight (HR 0.82, 95% CI 0.79–0.85), obesity (HR 0.81, 95% CI 0.78–0.85), and severe obesity categories (HR 0.76, 95% CI 0.70–0.82) had a lower risk of bronchiectasis development than the normal weight population. Stratified analyses by sex and age group in this subgroup also showed similar results (*p* for trend <0.01 for both). The effect of being underweight on bronchiectasis development was more robust in males than in females and in the older age group than in the younger age group (*p* for interaction <0.01 for both).

## 4. Discussion

This population-based, longitudinal, cohort study assessed the risk of bronchiectasis development according to BMI among young adults. The overall incidence of bronchiectasis was 0.4% for approximately a 7-year follow-up duration, and being underweight was found to be a novel risk factor for bronchiectasis development in young adults. BMI had interactions with sex and age in terms of bronchiectasis development; the effect of being underweight was more significant in males and older (30–39 years) individuals than females and younger (20–29 years) individuals, respectively.

Despite the well-known association between low BMI and worse clinical outcomes in patients with bronchiectasis [11], the association between BMI and susceptibility to bronchiectasis development has not been well elucidated. To the best of our knowledge, this study is the first to comprehensively evaluate the association between the BMI category and the risk of incident bronchiectasis in young adults. Specifically, we focused on young adults because they have relatively fewer comorbidities that could potentially contribute to a low BMI and bronchiectasis development. In particular, it is well known that bronchiectasis-related comorbidities (e.g., TB, NTM disease, inflammatory bowel disease, immunodeficiency, etc.) are closely related to low BMI [17,18,19]. Thus, those comorbidities could serve as confounding factors when interpreting the association between being underweight and the development of bronchiectasis. To minimize the effect of potential confounders, we further analyzed the relationship between BMI and bronchiectasis after excluding individuals with these diseases from the analysis and confirmed the negative correlation between BMI and risk for bronchiectasis development.

One possible explanation for the higher risk of bronchiectasis in subjects with lower BMI is increased inflammatory activity in underweight subjects. Being underweight indicates nutritional deficiencies and is related to increased pulmonary inflammation and free neutrophil elastase activity in the lungs [20], which has been suggested to underlie the association between poor nutritional status and worse lung disease. Numerous conditions in underweight individuals such as chronic energy deficiency, frequent pulmonary inflammation, increased oxidative stress, and altered body composition may contribute to the development of bronchiectasis [21,22]. Further research is warranted to elucidate the exact mechanisms underlying the association between being underweight and bronchiectasis development. Low BMI may also reflect childhood infections and growth environments, which can affect the development of bronchiectasis. However, we could not include childhood infections or growth environments as confounding variables in our analyses of the association between BMI and the occurrence of bronchiectasis, due to the lack of this data in the NHIS database. However, we addressed this issue by adjusting our analyses for income level and area of residence, as both of these factors reflect socioeconomic status [23].

We found that the effect of being underweight on the development of bronchiectasis was greater in males and older (30–39 years) individuals than females and younger (20–29 years) individuals, respectively. There are several potential explanations as to why the effect of being underweight was more significant in males than females. The sex disparity in chronic lung diseases, including bronchiectasis, has been suggested to be related to differences in genetics and sex hormones (types and concentrations) [24]. For example, bronchiectasis develops earlier in females than in males, while the presentation of bronchiectasis is more common in males than in females in old age groups [24]. These data suggest that male sex hormones may have a more protective effect than female sex hormones on the occurrence of bronchiectasis. Furthermore, BMI is known to be positively correlated with female hormone levels [25] but negatively correlated with male hormone levels [26]. Thus, complex interactions among BMI, sex, and sex hormones may explain the different effects of being underweight on the development of bronchiectasis by sex. Meanwhile, the reasons why the effect of being underweight on the occurrence of bronchiectasis is more profound in older individuals than younger individuals are not well known. Although it is known that the levels of sex hormones decrease by age [27], the changes might be small as our study population was relatively young. Thus, the change in the levels of sex hormones by aging on the development of bronchiectasis might be minimal in our population. Future studies are needed to provide more detailed information on the complex association between BMI, age, sex, and the development of bronchiectasis.

The major strength of this study is that it is the first longitudinal study to investigate the impact of being underweight on bronchiectasis development in young adults. In addition, we focused on the young population to minimize the effect of comorbidities as potential confounders related to low BMI. Nonetheless, this study had several limitations that should be acknowledged. First, the diagnosis of bronchiectasis was based on the physician’s diagnosis. Thus, there might have been under- or over-estimation of bronchiectasis. Second, there were no data on lung function and childhood pulmonary infection, which might be associated with the occurrence of bronchiectasis. Third, this study lacked detailed medication information. Thus, we could not evaluate the impact of medication (e.g., antibiotics including macrolide, steroids, bronchodilators, etc.) on the diagnosis of bronchiectasis. Fourth, as our study was performed with Koreans, our findings may not apply to other ethnic groups and populations.

## 5. Conclusions

This large, nationwide, longitudinal study demonstrated that being underweight may be a novel risk factor for bronchiectasis development in young adults. The effect of being underweight on bronchiectasis development was more robust in males and older (30–39 years) individuals than in females and younger (20–29 years) individuals, respectively. Future research is needed to confirm our findings regarding the adverse impact of being underweight on bronchiectasis development.

## Figures and Tables

**Figure 1 nutrients-13-03206-f001:**
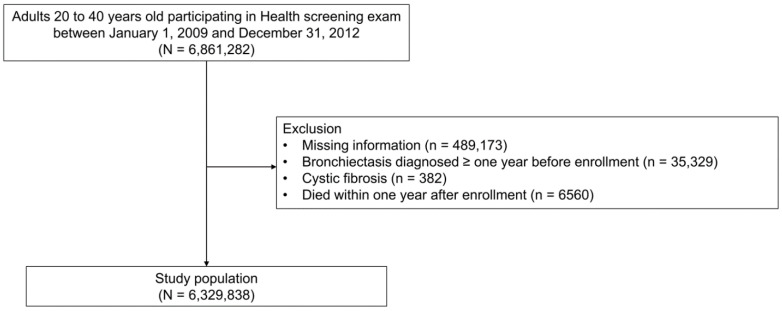
Flow chart of the study population.

**Figure 2 nutrients-13-03206-f002:**
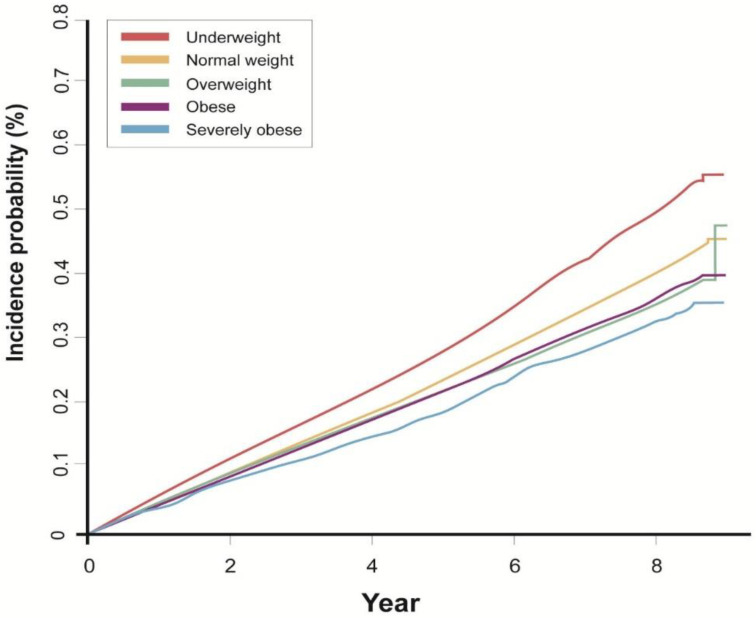
Cumulative incidence of bronchiectasis according to body mass index category (%).

**Figure 3 nutrients-13-03206-f003:**
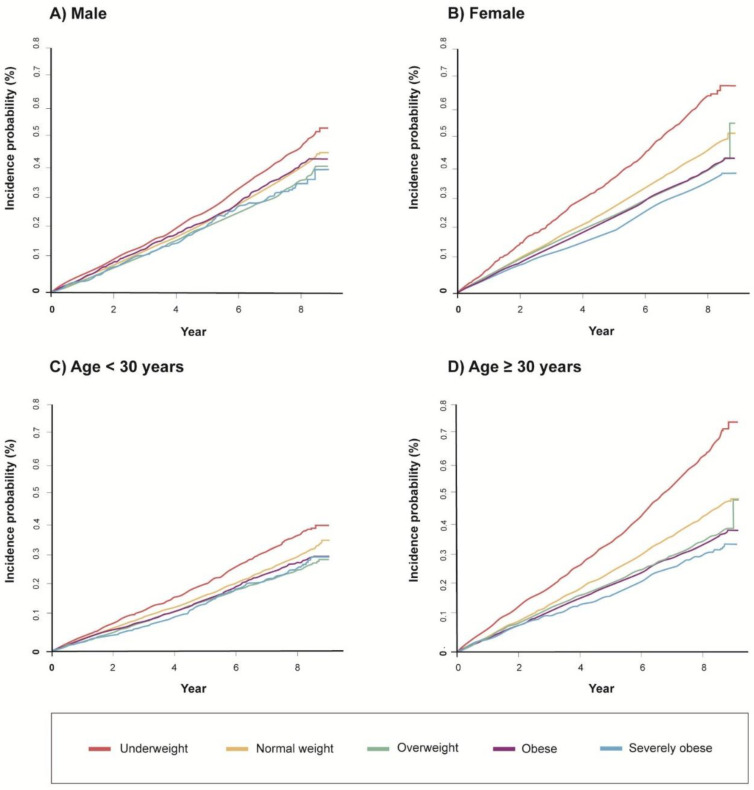
Cumulative incidence of bronchiectasis according to sex and age (%). (**A**) males, (**B**) females, (**C**) age <30 years, (**D**) age ≥30 years.

**Table 1 nutrients-13-03206-t001:** Baseline characteristics of the study population according to body mass index.

		Baseline BMI Level (kg/m^2^)					
	Total (n = 6,329,838)	Underweight (<18.5) (n = 479,506)	Normal Weight (18.5–22.9) (n = 2,953,857)	Overweight (23–24.9) (n = 1,215,973)	Obesity (25–29.9) (n = 1,409,751)	Severe Obesity (≥30) (n = 270,751)	*p*
Age, years	30.9 ± 5.0	28.6 ± 4.8	30.2 ± 5.1	31.6 ± 4.8	32.2 ± 4.6	31.4 ± 4.7	<0.01
<30	2,674,626 (42.3)	297,537 (62.1)	1,417,920 (48.0)	433,985 (35.7)	428,880 (30.4)	96,304 (35.6)	<0.01
≥30	3,655,212 (57.8)	181,969 (38.0)	1,535,937 (52.0)	781,988 (64.3)	980,871 (69.6)	174,447 (64.4)	
Sex							<0.01
Male	3,754,667 (59.3)	99,358 (20.7)	1,365,668 (46.2)	915,572 (75.3)	1,164,300 (82.6)	209,769 (77.5)	
Female	2,575,171 (40.7)	380,148 (79.3)	1,588,189 (53.8)	300,401 (24.7)	245,451 (17.4)	60,982 (22.5)	
Smoking status							<0.01
Never-smoker	346,2831 (54.7)	371,396 (77.5)	1,885,083 (63.8)	557,099 (45.8)	542,872 (38.5)	106,381 (39.3)	
Ex-smoker	658,223 (10.4)	21,415 (4.5)	239,078 (8.1)	163,642 (13.5)	201,822 (14.3)	32,266 (11.9)	
Current smoker	2,208,784 (34.9)	866,95 (18.1)	829,696 (28.1)	495,232 (40.7)	665,057 (47.2)	132,104 (48.8)	
Alcohol consumption							<0.01
None	2,382,737 (37.6)	244,670 (51.0)	1,236,249 (41.9)	393,772 (32.4)	417,787 (29.6)	90,259 (33.3)	
Mild	3,386,154 (53.5)	218,856 (45.6)	1,530,386 (51.8)	692,760 (57.0)	802,339 (56.9)	141,813 (52.4)	
Heavy	560,947 (8.9)	15,980 (3.3)	187,222 (6.3)	129,441 (10.7)	189,625 (13.5)	38,679 (14.3)	
Regular exercise							<0.01
No	5,515,870 (87.1)	447,998 (93.4)	2,615,229 (88.5)	1,033,135 (85.0)	1,190,658 (84.5)	228,850 (84.5)	
Yes	813,968 (12.9)	31,508 (6.6)	338,628 (11.5)	182,838 (15.0)	219,093 (15.5)	41,901 (15.5)	
Low income							<0.01
No	5,235,442 (84.1)	389,698 (81.3)	2,445,014 (82.8)	1,043,529 (85.8)	1,221,682 (86.7)	225,519 (83.3)	
Yes	1,004,396 (15.9)	89,808 (18.7)	508,843 (17.2)	172,444 (14.2)	188,069 (13.3)	45,232 (16.7)	
Residence							<0.01
Rural	3,305,465 (52.2)	236,233 (49.3)	1,532,027 (51.9)	643,974 (53.0)	749,018 (53.1)	144,213 (53.3)	
Urban	3,024,373 (47.8)	243,273 (50.7)	1,421,830 (48.1)	571,999 (47.0)	660,733 (46.9)	126,538 (46.7)	
Number of hospital visits	3.6 ± 6.5	3.8 ± 6.7	3.7 ± 6.5	3.5 ± 6.5	3.5 ± 6.5	3.4 ± 6.6	<0.01
Admission	0.0 ± 0.3	0.0 ± 0.2	0.0 ± 0.3	0.1 ± 0.3	0.1 ± 0.3	0.1 ± 0.3	<0.01
Outpatient	3.6 ± 6.5	3.8 ± 6.6	3.6 ± 6.5	3.5 ± 6.4	3.5 ± 6.4	3.3 ± 6.6	<0.01
Comorbidities							
DM	123,254 (2.0)	3034 (0.6)	28,824 (1.0)	22,209 (1.8)	48,538 (3.4)	20,649 (7.6)	<0.01
CKD or ESRD	2345 (0)	195 (0)	1023 (0)	410 (0)	551 (0)	166 (0.1)	<0.01
GERD	679,531 (10.7)	63,492 (13.2)	330,006 (11.2)	119,921 (9.9)	139,312 (9.9)	26,800 (9.9)	<0.01
Respiratory disease	331,045 (5.2)	27,675 (5.8)	157,277 (5.3)	59,836 (4.9)	70,746 (5.0)	15,511 (5.7)	<0.01
Asthma	328,883 (5.2)	27,351 (5.7)	156,042 (5.3)	59,521 (4.9)	70,497 (5.0)	15,472 (5.7)	<0.01
TB	2438 (0)	347 (0.1)	1393 (0.1)	364 (0)	294 (0)	40 (0)	<0.01
NTM disease	106 (0)	23 (0)	47 (0)	19 (0)	11 (0)	6 (0)	<0.01
Others							
Connective tissue disease	45,055 (0.7)	3824 (0.8)	21,274 (0.7)	8068 (0.7)	9785 (0.7)	2104 (0.8)	<0.01
Solid cancer	12,652 (0.2)	1198 (0.3)	6748 (0.2)	2076 (0.2)	2168 (0.2)	462 (0.2)	<0.01
Hematologic malignancy	14 (0)	1 (0)	8 (0)	2 (0)	3 (0)	0 (0)	0.89
Transplantation	54 (0)	6 (0)	26 (0)	10 (0)	11 (0)	1 (0)	0.78
HIV/AIDS	446 (0)	22 (0)	219 (0)	104 (0)	86 (0)	15 (0)	0.02
Immunodeficiency	295 (0)	35 (0)	148 (0)	49 (0)	48 (0)	15 (0)	0.01
Inflammatory bowel disease	7983 (0.1)	840 (0.2)	3951 (0.1)	1474 (0.1)	1482 (0.1)	236 (0.1)	<0.01

Abbreviations: BMI, body mass index; DM, diabetes mellitus; CKD, chronic kidney disease; ESRD, end-stage renal disease; HIV, human immunodeficiency virus; AIDS, acquired immunodeficiency syndrome; GERD, gastroesophageal reflux disease; TB, tuberculosis; NTM, nontuberculous mycobacteria.

**Table 2 nutrients-13-03206-t002:** Risk of bronchiectasis according to BMI category.

						HR (95% CI)		
	BMI	No at Risk	Incident Bronchiectasis	Follow-Up Duration (PY)	IR per 1000 PY	Model 1	Model 2	Model 3
Overall	<18.5	479,506	2281	3,499,101	0.652	1.24 (1.19–1.30)	1.36 (1.30–1.43)	1.36 (1.30–1.42)
	18.5–22.9	2953,857	11,425	21,721,387	0.526	1 (reference)	1 (reference)	1 (reference)
	23–24.9	1215,973	4226	9,010,885	0.469	0.89 (0.86–0.92)	0.83 (0.80–0.86)	0.83 (0.80–0.86)
	25–29.9	1409,751	5001	10,434,930	0.479	0.91 (0.88–0.94)	0.82 (0.80–0.85)	0.82 (0.79–0.85)
	≥30	270,751	871	1,965,539	0.443	0.84 (0.79–0.90)	0.79 (0.74–0.85)	0.79 (0.73–0.84)
	*p* for trend					<0.01	<0.01	<0.01
	*p*					<0.01	<0.01	<0.01
Sex								
Male	<18.5	99,358	608	731,219	0.831	1.54 (1.42–1.68)	1.64 (1.50–1.78)	1.63 (1.50–1.77)
	18.5–22.9	1,365,668	5475	10,149,938	0.539	1 (reference)	1 (reference)	1 (reference)
	23–24.9	915,572	3194	6,831,295	0.468	0.87 (0.83–0.91)	0.82 (0.78–0.85)	0.82 (0.78–0.85)
	25–29.9	1,164,300	4071	8,665,586	0.470	0.87 (0.84–0.91)	0.80 (0.77–0.83)	0.79 (0.76–0.83)
	≥30	209,769	651	1,532,268	0.425	0.79 (0.73–0.86)	0.75 (0.69–0.81)	0.74 (0.69–0.81)
	*p* for trend					<0.01	<0.01	<0.01
	*p*					<0.01	<0.01	<0.01
Female	<18.5	380,148	1673	2,767,882	0.604	1.18 (1.11–1.24)	1.30 (1.23–1.37)	1.30 (1.23–1.37)
	18.5-22.9	1,588,189	5950	11,571,449	0.514	1 (reference)	1 (reference)	1 (reference)
	23-24.9	300,401	1032	2,179,590	0.473	0.92 (0.86–0.98)	0.85 (0.80–0.91)	0.85 (0.79–0.91)
	25-29.9	245,451	930	1,769,344	0.526	1.02 (0.96–1.10)	0.93 (0.87–1.00)	0.92 (0.86–0.99)
	≥30	60,982	220	433,271	0.508	0.99 (0.87–1.13)	0.94 (0.82–1.07)	0.92 (0.80–1.05)
	*p* for trend					<0.01	<0.01	<0.01
	*p*					<0.01	<0.01	<0.01
	*p* for interaction					<0.01	<0.01	<0.01
Age, years								
<30	<18.5	297,537	981	2,163,489	0.453	1.21 (1.13–1.30)	1.21 (1.12–1.30)	1.21 (1.12–1.29)
	18.5–22.9	1,417,920	3868	10,294,737	0.376	1 (reference)	1 (reference)	1 (reference)
	23–24.9	433,985	1019	3,143,072	0.324	0.86 (0.81–0.93)	0.85 (0.80–0.92)	0.85 (0.79–0.92)
	25–29.9	428,880	1060	3,099,629	0.342	0.91 (0.85–0.98)	0.89 (0.83–0.96)	0.89 (0.83–0.95)
	≥30	96,304	238	686,836	0.347	0.92 (0.81–1.05)	0.91 (0.80–1.04)	0.90 (0.79–1.03)
	*p* for trend					<0.01	<0.01	<0.01
	*p*					<0.01	<0.01	<0.01
≥30	<18.5	181,969	1300	1,335,612	0.973	1.47 (1.39–1.56)	1.51 (1.42–1.60)	1.51 (1.42–1.60)
	18.5–22.9	1,535,937	7557	11,426,650	0.661	1 (reference)	1 (reference)	1 (reference)
	23–24.9	781,988	3207	5,867,813	0.547	0.83 (0.79–0.86)	0.82 (0.79–0.86)	0.82 (0.79–0.86)
	25–29.9	980,871	3941	7,335,301	0.537	0.81 (0.78–0.84)	0.81 (0.78–0.84)	0.80 (0.77–0.84)
	≥30	174,447	633	1,278,703	0.495	0.75 (0.69–0.81)	0.76 (0.70–0.82)	0.75 (0.69–0.81)
	*p* for trend					<0.01	<0.01	<0.01
	*p*					<0.01	<0.01	<0.01
	*p* for interaction					<0.01	<0.01	<0.01

Model 1 is unadjusted; Model 2 was adjusted for age, sex, smoking status, alcohol consumption, regular exercise, income low, place, and number of hospital visits; Model 3 was additionally adjusted for respiratory disease, connective tissue disease, solid cancer, hematologic malignancy, transplantation, immunodeficiency, inflammatory bowel disease, and HIV and AIDS. Abbreviations: BMI, body mass index; HR, hazard ratio; CI, confidence interval; PY, person-years; IR, incidence rate; HIV, human immunodeficiency virus; AIDS, acquired immunodeficiency syndrome.

**Table 3 nutrients-13-03206-t003:** Subgroup analysis of the risk of bronchiectasis according to BMI category.

						HR (95% CI)	
	BMI (kg/m^2^)	No at Risk	Incident Bronchiectasis	Follow-Up Duration (PY)	IR per 1000 PY	Model 1	Model 2
Overall	<18.5	435,806	1916	3,184,697	0.602	1.23 (1.17–1.29)	1.35 (1.29–1.42)
	18.5–22.9	2,706,903	9770	19,928,975	0.490	1 (reference)	1 (reference)
	23–24.9	1,121,922	3622	8,322,146	0.35	0.89 (0.85–0.92)	0.82 (0.79–0.85)
	25–29.9	1,299,192	4287	9,626,359	0.445	0.91 (0.88–0.94)	0.81 (0.78–0.85)
	≥30	247,060	714	1,796,265	0.397	0.81 (0.75–0.88)	0.76 (0.70–0.82)
	*p* for trend					<0.01	<0.01
	*p*					<0.01	<0.01
Sex							
Male	<18.5	93,246	528	866,557	0.769	1.51 (1.38–1.66)	1.60 (1.46–1.75)
	18.5–22.9	1,280,799	4842	9,523,140	0.508	1 (reference)	1 (reference)
	23–24.9	854,399	2793	6,377,645	0.438	0.86 (0.82–0.90)	0.81 (0.78–0.85)
	25–29.9	1,082,933	3539	8,064,567	0.439	0.86 (0.83–0.90)	0.79 (0.76–0.83)
	≥30	194,485	555	1,421,622	0.390	0.77 (0.70–0.84)	0.73 (0.67–0.80)
	*p* for trend					<0.01	<0.01
	*p*					<0.01	<0.01
Female	<18.5	342,560	1388	2,498,140	0.556	1.17 (1.11–1.25)	1.29 (1.21–1.37)
	18.5–22.9	1,426,104	4928	10,405,835	0.474	1 (reference)	1 (reference)
	23–24.9	267,523	829	1,944,501	0.426	0.90 (0.84–0.97)	0.83 (0.71–0.89)
	25–29.9	216,259	748	1,561,793	0.479	1.01 (0.94–1.09)	0.92 (0.85–1.00)
	≥30	52,575	159	374,643	0.424	0.90 (0.77–1.05)	0.85 (0.73–1.00)
	*p* for trend					<0.01	<0.01
	*p*					<0.01	<0.01
	*p* for interaction					<0.01	<0.01
Age, years							
<30	<18.5	273,143	851	2,498,140	0.428	1.22 (1.13–1.31)	1.22 (1.13–1.32)
	18.5–22.9	1,316,145	3370	10,405,835	0.352	1 (reference)	1 (reference)
	23–24.9	406,352	898	1,944,501	0.305	0.87 (0.80–0.93)	0.85 (0.79–0.91)
	25–29.9	401,092	954	1,561,793	0.329	0.93 (0.87–1.00)	0.90 (0.84–0.97)
	≥30	89,044	199	374,643	0.313	0.89 (0.77–1.03)	0.87 (0.75–1.01)
	*p* for trend					<0.01	<0.01
	*p*					<0.01	<0.01
≥30	<18.5	162,663	1065	1,196,246	0.890	1.44 (1.35–1.54)	1.48 (1.39–1.58)
	18.5–22.9	1,390,758	6400	10,363,190	0.618	1 (reference)	1 (reference)
	23–24.9	715,570	2724	5,376,246	0.507	0.82 (0.78–0.86)	0.81 (0.77–0.85)
	25–29.9	898,100	3333	6,724,833	0.496	0.80 (0.77–0.84)	0.79 (0.76–0.82)
	≥30	158,016	515	1,160,403	0.444	0.72 (0.66–0.79)	0.72 (0.66–0.79)
	*p* for trend					<0.01	<0.01
	*p*					<0.01	<0.01
	*p* for interaction					<0.01	<0.01

Model 1 is unadjusted; Model 2 is adjusted for age, sex, smoking status, alcohol consumption, regular exercise, income low, place, and the number of hospital visits. Abbreviations: BMI, body mass index; HR, hazard ratio; CI, confidence interval; PY, person-years; IR, incidence rate.

## Data Availability

Data supporting reported results are available upon reasonable request and in accordance with the ethical principles.

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
