# Peer review of "Being Underweight Increases the Risk of Non-Cystic Fibrosis Bronchiectasis in the Young Population: A Nationwide Population-Based Study"

_nutrients, 2021, doi:10.3390/nu13093206_

Round 1

Reviewer 1 Report

This study was conducted to investigate the effect of BMI on the development of bronchiectasis in young adults and was conducted with a very clear and appropriate design.

I have some minor comments.

1.Has the Korean National Health Insurance Service (NHIS) data used in this study been subjected to any validation studies, including validity of disease names? If so, please show them in the paper.

2.Repeated airway infections and prolonged use of antimicrobial agents such as macrolides are said to affect the onset of bronchiectasis, but is it possible to adjust for these factors?

Author Response

Responses to Comments

## Response to Reviewer 1’s comments

General comments

This study was conducted to investigate the effect of BMI on the development of bronchiectasis in young adults and was conducted with a very clear and appropriate design.
Response. Thank you for your positive comments. Regarding the concerns raised by the reviewer, we provided point-by-point responses as below.

Specific comments

Comment 1 (C1). Has the Korean National Health Insurance Service (NHIS) data used in this study been subjected to any validation studies, including validity of disease names? If so, please show them in the paper.
Response 1 (R1). Unfortunately, there are no definite validation studies for the definition of bronchiectasis using NHIS data. However, our team previously reported the prevalence of bronchiectasis in South Korea, which was correspondent to 496 per 100,000 population [1]. To validate the prevalence of bronchiectasis using other methods, we further evaluated the dataset of the Korean National Health and Nutrition Examination Survey (NHANES). The Korea NHANES is a cross-sectional, nationally representative survey of the non-institutionalized South Korean population conducted by the Korean Ministry of Health and Welfare using a stratified, multistage clustered probability sampling design. The 2007–2009 survey included the question, “Have you ever been diagnosed with bronchiectasis by physicians?” When we analyzed these data, the prevalence of physician-diagnosed bronchiectasis in subjects ≥20 years of age was 423/100,000 population (77/18,210) [2], which is comparable with the prevalence diagnosed with ICD-10 code (J47) in our previous study [1].

References

  1. Choi, H.; Yang, B.; Nam, H.; Kyoung, D.S.; Sim, Y.S.; Park, H.Y.; Lee, J.S.; Lee, S.W.; Oh, Y.M.; Ra, S.W.; et al. Population-based prevalence of bronchiectasis and associated comorbidities in south korea. Eur Respir J 2019, 54, 1900194.
  2. Yang, B.; Choi, H.; Lim, J.H.; Park, H.Y.; Kang, D.; Cho, J.; Lee, J.S.; Lee, S.W.; Oh, Y.M.; Moon, J.Y.; et al. The disease burden of bronchiectasis in comparison with chronic obstructive pulmonary disease: A national database study in korea. Ann Transl Med 2019, 7, 770.

C2. Repeated airway infections and prolonged use of antimicrobial agents such as macrolides are said to affect the onset of bronchiectasis, but is it possible to adjust for these factors?

R2. Thanks for the helpful comments. Unfortunately, drug codes are not currently available in our dataset since the Korean National Health Insurance Service (NHIS) provided the dataset without drug codes. The Korean NHIS does not provide all data, but only the requested data based on the hypothesis of the researchers. At this stage, an additional request for data is not permitted by the NHIS. As we did not consider repeated airway infections and prolonged use of antimicrobial agents such as macrolides in our analyses, we added this as a limitation in the Discussion section of the revised manuscript (page 12, lines 332–334).

Third, this study lacked detailed medication information. Thus, we could not evaluate the impact of medication (e.g., antibiotics including macrolide, steroids, bronchodilators, etc.) on the diagnosis of bronchiectasis”

Reviewer 2 Report

The paper entitled” Underweight increases the risk of non-cystic fibrosis bronchiectasis in the young population: a nationwide population-based study” was reviewed. The authors collected a national-wide study from the Korean National Health Insurance Service database 2009–2012 to enroll 6,329,838 individuals aged 20–40 years. The patients were followed up until the date of the diagnosis of bronchiectasis, death, or 31 December 2018. They evaluated the incidence and risk of bronchiectasis according to the BMI category. They found that 23,804 (0.4%) subjects developed bronchiectasis during a mean of 7.4 years of follow-up, and underweight (BMI < 18.5 kg/m2) was an independent risk factor for the development of bronchiectasis (HR=1.24), and the effect of underweight on the development of bronchiectasis was more evident in males and older individuals (30–40 years).

   The paper was well written, with a large, nationwide, longitudinal study to survey the possible impact of underweight on the incidences of young individuals with bronchiectasis. I had some comments:

  1. As the authors described, the diagnosis of bronchiectasis was based on the physician’s diagnosis, without any prove by CXR or lung function test, and it might be due to the diagnosis was just for need for medication. Can the authors show some detail about the use of medication on the impact of diagnosis? As antibiotics, steroids, bronchodilator, aminophylline or antihistamine?
  2. What about the impact of smoking on bronchiectasis in your study? Can the authors display more information or comments?

Author Response

## Response to Reviewer 2’s comments

General comments

The paper entitled” Underweight increases the risk of non-cystic fibrosis bronchiectasis in the young population: a nationwide population-based study” was reviewed. The authors collected a national-wide study from the Korean National Health Insurance Service database 2009–2012 to enroll 6,329,838 individuals aged 20–40 years. The patients were followed up until the date of the diagnosis of bronchiectasis, death, or 31 December 2018. They evaluated the incidence and risk of bronchiectasis according to the BMI category. They found that 23,804 (0.4%) subjects developed bronchiectasis during a mean of 7.4 years of follow-up, and underweight (BMI < 18.5 kg/m2) was an independent risk factor for the development of bronchiectasis (HR=1.24), and the effect of underweight on the development of bronchiectasis was more evident in males and older individuals (30–40 years).
The paper was well written, with a large, nationwide, longitudinal study to survey the possible impact of underweight on the incidences of young individuals with bronchiectasis. I had some comments:

Response. Thank you for your positive comments. Regarding the concerns raised by the reviewer, we provided point-by-point responses as below.

Specific comments

C1. As the authors described, the diagnosis of bronchiectasis was based on the physician’s diagnosis, without any prove by CXR or lung function test, and it might be due to the diagnosis was just for need for medication. Can the authors show some detail about the use of medication on the impact of diagnosis? As antibiotics, steroids, bronchodilator, aminophylline or antihistamine?
R1. Thanks for the helpful comments. As the reviewer commented, the absence of CXR findings or lung function tests is an important limitation to our study. And there is a possibility that the physician made a diagnosis of bronchiectasis to prescribe medication. Thus, we agree with the reviewer that the impact of some details about medication use on diagnosis would be very interesting. However, unfortunately, drug codes are not currently available in our dataset since the Korean National Health Insurance Service (NHIS) provided the dataset without drug codes. The Korean NHIS does not provide all data, but only the requested data based on the research hypothesis of the researchers. At this stage, an additional request for data is not permitted by the NHIS. As we did not consider the use of medication in our analysis, we have added this as a limitation in the Discussion section of the revised manuscript (page 12, lines 332–334).

Third, this study lacked detailed medication information. Thus, we could not evaluate the impact of medication (e.g., antibiotics including macrolide, steroids, bronchodilators, etc.) on the diagnosis of bronchiectasis.”

C2. What about the impact of smoking on bronchiectasis in your study? Can the authors display more information or comments?
R2. Thanks for the helpful comment. We fully agree with your opinion that the impact of smoking on bronchiectasis is also important. However, the topic is out of the scope of our study purpose (the association between BMI and bronchiectasis), and to be honest with you, we are researching this topic. In our preliminary analyses, during about a 10-year follow-up duration, ex-smokers and current smokers had 1.07 (95% CI, 1.02–1.12) and 1.06 (95% CI, 1.02–1.10) times increased risk of bronchiectasis than never smokers, respectively. As we are investigating this topic in independent research, we hope the reviewer understands that we could not add information on the impact of smoking on bronchiectasis to this manuscript.
